# The Distribution of *Puumala orthohantavirus* Genome Variants Correlates with the Regional Landscapes in the Trans-Kama Area of the Republic of Tatarstan

**DOI:** 10.3390/pathogens10091169

**Published:** 2021-09-10

**Authors:** Yuriy N. Davidyuk, Emmanuel Kabwe, Anton F. Shamsutdinov, Anna V. Knyazeva, Ekaterina V. Martynova, Ruzilya K. Ismagilova, Vladimir A. Trifonov, Tatiana A. Savitskaya, Guzel S. Isaeva, Richard A. Urbanowicz, Svetlana F. Khaiboullina, Albert A. Rizvanov, Sergey P. Morzunov

**Affiliations:** 1OpenLab “Gene and Cell Technologies”, Institute of Fundamental Medicine and Biology, Kazan Federal University, 420008 Kazan, Russia; davi.djuk@mail.ru (Y.N.D.); emmanuelkabwe@ymail.com (E.K.); shamsutdinov2006@yandex.com (A.F.S.); cnyaz.anna@yandex.ru (A.V.K.); ignietferro.venivedivici@gmail.com (E.V.M.); rizvanov@gmail.com (A.A.R.); 2Kazan Research Institute of Epidemiology and Microbiology, 420012 Kazan, Russia; vatrifonov@mail.ru (V.A.T.); tatasav777@mail.ru (T.A.S.); guisaeva@rambler.ru (G.S.I.); 3OpenLab “Omics Technology”, Institute of Fundamental Medicine and Biology, Kazan Federal University, 420008 Kazan, Russia; ruz-ismagilova@yandex.ru; 4Medical Academy of the Ministry of Health of the Russian Federation, 420012 Kazan, Russia; 5Department of Infection Biology and Microbiomes, Institute of Infection, Veterinary and Ecological Sciences, University of Liverpool, Liverpool L3 5RF, UK; Richard.Urbanowicz@liverpool.ac.uk; 6Department of Pathology, University of Nevada, Reno, NV 89557, USA

**Keywords:** *Puumala orthohantavirus*, genetic diversity, phylogenetic analysis, *Myodes glareolus*, Trans-Kama area, Republic of Tatarstan

## Abstract

In the European part of Russia, the highest number of hemorrhagic fever with renal syndrome (HFRS) cases are registered in the Volga Federal District (VFD), which includes the Republic of Tatarstan (RT). *Puumala orthohantavirus* (PUUV) is the main causative agent of HFRS identified in the RT. The goal of the current study is to analyze the genetic variations of the PUUV strains and possible presence of chimeric and reassortant variants among the PUUV strains circulating in bank vole populations in the Trans-Kama area of the RT. Complete S segment CDS as well as partial M and L segment coding nucleotide sequences were obtained from 40 PUUV-positive bank voles and used for the analysis. We found that all PUUV strains belonged to RUS genetic lineage and clustered in two subclades corresponding to the Western and Eastern Trans-Kama geographic areas. PUUV strains from Western Trans-Kama were related to the previously identified strain from Teteevo in the Pre-Kama area. It can be suggested that the PUUV strains were introduced to the Teteevo area as a result of the bank voles’ migration from Western Trans-Kama. It also appears that physical obstacles, including rivers, could be overcome by migrating rodents under favorable circumstances. Based on results of the comparative and phylogenetic analyses, we propose that bank vole distribution in the Trans-Kama area occurred upstream along the river valleys, and that watersheds could act as barriers for migrations. As a result, the diverged PUUV strains could be formed in closely located populations. In times of extensive bank vole population growth, happening every 3–4 years, some regions of watersheds may become open for contact between individual rodents from neighboring populations, leading to an exchange of the genetic material between divergent PUUV strains.

## 1. Introduction

Orthohantaviruses, which belong to the genus *Orthohantavirus*, family *Hantaviridae*, order *Bunyavirales,* are zoonotic pathogens circulating in small mammals in the Americas and Eurasia [1]. Orthohantaviruses are divided into two main groups, each associated with distinct clinical symptoms in human infections [2]. In the Americas, orthohantaviruses cause hantavirus pulmonary syndrome, whilst nephropathia epidemica (NE), a mild form of hemorrhagic fever with renal syndrome (HFRS), is documented in Eurasia [3]. Orthohantaviruses are endemic in Europe, representing a constant health care threat. Annually, massive outbreaks of HFRS are reported in Fennoscandia, Central and Western Europe, as well as in Russia [4,5,6]. Severe cases, some with lethal outcome, have been reported, in particular, in Switzerland, Russia, and Finland [7,8,9]. It is believed that the mode of orthohantaviruses transmission is by inhaling aerosolized excreta contaminated with virus released by infected rodents [10].

In the European part of Russia, the highest number of HFRS cases is registered in the Volga Federal District (VFD), which includes the Republic of Tatarstan (RT). The RT is amongst the regions of the Russian Federation (RF) with the highest annual number of HFRS cases [11]. In 2019 alone, more than 700 cases of HFRS were reported in the RT [6]. With an increasing number of HFRS cases in this region, the RT is ranked fourth with respect to the prevalence of this zoonosis infectious disease after Udmurtia, Bashkiria, and Mordovia in the VFD of the RF. *Puumala orthohantavirus* (PUUV) is one of the commonly identified causative agents of human HFRS in this region and causes NE, a mild form of the disease with a fatality rate up to 0.4% [10]. In nature, PUUV circulates in bank voles (*Myodes glareolus*) as its primary reservoir [3,11,12].

The genome of PUUV is tripartite and comprised of small (S), medium (M), and large (L), single-stranded, negative-polarity RNA segments. The S segment is 1830 nucleotides (nt)-long and codes for the nucleocapsid (N) protein of 433 amino acids (aa) [13]. In some orthohantaviruses, the S segment codes for another putative nonstructural protein of 90 aa in overlapping open reading frame (ORF) [14]. The M segment is 3682 nt-long, coding for the precursor of the envelope glycoproteins (Gn and Gc) of 1148 aa [15]. The L segment codes for the RNA-dependent RNA polymerase (RdRp) of 2156 aa and is 6550 nt-long [16].

Currently, eight PUUV genetic lineages have been identified in Eurasia, including (1) the Central European (CE) found in France, Belgium, Germany, Netherland, and Slovakia; (2) the Alpe-Adrian (ALAD) spread in Austria, Slovenia, Croatia, and Hungary; (3) the Danish (DAN) detected in Denmark; (4) the South-Scandinavian (S-SCA) found in Norway and Sweden (central and southern); (5) the North-Scandinavian (N-SCA) detected in northern Sweden and northwestern Finland; (6) the Finnish (FIN) spread in Finland, Russian Karelia, and western Siberia (Omsk region); (7) the Russian (RUS) found in pre-Ural Russia (this family has been found in Samara Oblast, the Republic of Udmurtia, RT, Republic of Bashkortostan, Republic of Mordovia, Ulyanovsk Oblast) and Baltic countries such as Estonia and Latvia; and (8) the recently discovered Latvian (LAT) endemic in Latvia, Lithuania, and Poland [17,18]. Though some of these PUUV genetic lineages are found in several countries, geographical clustering of the virus strains mainly reflects geographic distribution of the particular bank vole populations [10,17].

In our previous work, we investigated the distribution of PUUV genome variants in the geographic region of the RT, called the Pre-Kama area. We showed that the PUUV strains display significant genetic diversity even in closely located bank vole populations [19]. The goal of the current study is to analyze the genetic variations of the PUUV strains circulating in bank vole populations located in the Trans-Kama area of the RT. Specifically, we want to analyze possible presence of chimeric strains and reassortment events among the PUUV strains in rodent population.

## 2. Materials and Methods

### 2.1. Rodent Tissues Sampling

Rodents were captured in the Trans-Kama region of the RT during the period of 2015–2020 at 21 sites in an area of more than 20,000 km^2^. For unambiguous interpretation of the results, all trapping sites were given a number and an annotation such as S1 for site1. Geographic locations of rodent trapping sites are indicated in Figure 1. Frozen rodent lung tissue samples and information about trapping location were obtained from Federal Healthcare Institute “Centre for Hygiene and Epidemiology in the Republic of Tatarstan (Tatarstan)”.

### 2.2. RNA Extraction, cDNA Synthesis, and Polymerase Chain Reaction (PCR)

Total RNA was extracted from the lung tissues of small rodents using TRIzol Reagent (Invitrogen Life Technologies, Waltham, MA, USA), according to the manufacturer’s instructions. cDNA synthesis was performed using Thermo Scientific RevertAid Reverse Transcriptase (Thermo Fisher Scientific, Waltham, MA, USA) following the manufacturer’s recommendations. RT-PCR amplification was carried out using TaqPol polymerase (Evrogen, Moscow, Russia) as specified by the manufacturer. Forward and reverse primers used both for RT-PCRs amplification and sequencing were designed as described by Davidyuk et al. [19].

PCR amplicons were purified using Isolate II PCR and Gel Kit (Bioline, London, UK) and sequenced using ABI PRISM 310 Big Dye Terminator 3.1 sequencing kit (ABI, Waltham, MA, USA) per specifications of the manufacturer. Sequences were deposited in the GenBank database under the following accession numbers: MW484947-MW484950 and MW504212-MW504246 for complete coding sequence (CDS) of the S segment; MW498163-MW498198 for partial M segment; and MW498199-MW498238 for partial L segment.

### 2.3. Phylogenetic Analysis

Multiple nucleotide sequence alignment and phylogenetic analysis of the PUUV strains were conducted using the MegAlign program (Clustal W algorithm) from the DNASTAR software package Lasergene (DNASTAR, Madison, WI, USA; https://www.dnastar.com/ accessed on 20 June 2021) and MEGA v6.0, respectively [20]. Phylogenetic trees were inferred using Maximum Parsimony (MP) method incorporated in Mega v6.0. [20]. The bootstrap values calculated for 1000 replicates are presented as percentages and the values less than 70% were not shown in the tree. The tree is drawn with branch lengths reflecting the number of substitutions per site.

For comparison, nucleotide sequences of PUUV strains obtained in this work and from GenBank were used. The complete list of the sequences used for comparison is presented in the Appendix A. Sequences of *Tula orthohantavirus* AF164093, NC_005228, and NC_005226 for segments S, M, and L, respectively, were used as an outgroup.

## 3. Results

### 3.1. Screening of Rodents

A total of 162 bank vole lung tissue samples were used in our study. Viral RNA was detected in 59 samples (Table 1). Complete CDS S (1302 bp), partial M (1014 bp), and partial L (665 bp) segments were obtained from 40 out of 59 PUUV-positive samples (Appendix A). All PUUV strains identified were named to include their corresponding virus name, trapping region, sample number, and year (for example, PUUV/Aksarino/MG_108/2015). Further, for conciseness, we will use abbreviated notation—for instance, MG_108.

### 3.2. Molecular Analysis of the Trans-Kama PUUV Strains

#### 3.2.1. Comparison of the S Segment Nucleotide and N Protein Amino Acid Sequences of PUUV

Sequence analyses identified low divergence (less than 2%) in the S segments obtained from the same site (Figure 2, Appendix A). All the currently identified sequences from the Trans-Kama area (’Trans-Kama sequences’) formed 13 groups, mostly based on genetic distances and location of the trapping sites (Appendix A). Within most of the groups, the divergence did not exceed 0.2%. In groups 4 and 10, divergences ranged from zero to 1.4% and, within group 1, the diversity was higher, up to 2.0%. The intergroup sequence divergence was up to 6.0% (Figure 2, Appendix A). Analysis of these sequences and those previously published by Davidyuk et al. [19], as well as other PUUV sequences belonging to the RUS lineage (Samara_49/CG/2005, Puu/Kazan, CG1820, DTK/Ufa-97 strains), showed 2.6–6.9% divergence (Figure 3, Appendix A). Throughout this article, we will use the terms ‘Pre-Kama sequences’/or ‘Pre-Kama strains’, referring to previously identified strains in the Pre-Kama area. The sequence difference was higher, exceeding 15%, when Trans-Kama sequences were compared with those from FIN, CE, and N-SCA lineages (Figure 3, Appendix A). This divergence fell within the accepted range for PUUV sequence difference between distinct PUUV lineages. We concluded that all S segment sequences identified in this study (Trans-Kama sequences) belong to the RUS genetic lineage.

The analysis revealed limited differences in the deduced amino acid sequences of the N protein in the Trans-Kama strains, ranging from 0 to 0.7% between groups. Corresponding differences ranged from zero to 1.4% when compared with the RUS lineage strains (Appendix A). The V149I aa substitution was found in MG_108 and MG_109 Trans-Kama strains, and it appears to be specific for site S8. The R aa residue at position 242 was detected in all PUUV strains from the Trans-Kama area except for MG_1419 and MG_1992. Interestingly, this aa is also specific for the Samara strain (Figure 4).

#### 3.2.2. Comparison of the Partial M Segment Nucleotide and Glycoprotein Precursor Amino Acid Sequences of PUUV

The M segment sequences displayed nucleotide diversity similar to the S segment. They formed 13 groups (Appendix A), comprising the same strains as for the S segment mostly based on the divergence and the geographic location of the trapping sites (Figure 5, Appendix A). The nucleotide sequence divergence among samples within each group varied from zero to 2.6% (Figure 5, Appendix A). The variation between sequences from different groups was 1.0–7.0%, similar to the S segment values (Figure 5, Appendix A). When the Trans-Kama strains were compared with Pre-Kama strains and other strains of the RUS genetic lineage, nucleotide divergence was within the range of 3.2–9.3% (Figure 3, Appendix A). In contrast, the divergence between Trans-Kama strains and CG1820 and DTK/Ufa-97 strains (‘Bashkiria strains’) was much higher (14.9–18.0%). It is an interesting observation as Bashkiria strains also belong to the RUS genetic lineage. Similarly, higher differences were found between Trans-Kama strains and FIN lineage strains (16.4–19.8%). Divergence of the nucleotide sequences between Trans-Kama strains, CE, and N-SCA was within the range accepted for divergence between different genetic lineages (21.6–26.1%) (Figure 3, Appendix A).

Analysis of the aa sequences of the glycoprotein precursor fragments showed from zero to 0.6% difference within each group. Between groups, these aa sequences varied 0.0–2.1% (Appendix A). Comparing aa sequences between Trans-Kama and Pre-Kama, the divergence was even lower, 0.0–1.8%. Further, the comparison of aa sequences of these strains to other PUUV isolates from Russia revealed that the difference with Samara_49/CG/2005 and Puu/Kazan strains was 0.3–1.8%. Interestingly, aa sequences divergence between the PUUV strains in this study and Bashkiria strains from Russia was 1.2–3.9% and close to aa sequence difference with FIN lineage strains (2.7–3.9%). However, higher differences (6.7–9.6%) were found when Trans-Kama aa sequences were compared with other PUUV genetic lineages (Appendix A). Moreover, analysis of the Trans-Kama aa sequences revealed five unique amino acid substitutions. In particular, aa substitutions I577V and I677V were found only in group 9, while substitutions A792V, K798R, and N710S were specific for groups 2, 2b, and 3, respectively (Figure 4).

#### 3.2.3. Comparison of the Partial L Segment Nucleotide and RdRp Amino Acid Sequences of PUUV

Groups 1, 2b, 4, 6, and 8–13, formed based on the results of an analysis of the L segment sequences, included the same strains as the corresponding groups for S and M segments. However, strains MG_1555, MG1199, and MG_1998 fell into different groups—2a, 5, and 7, respectively (Appendix A). Analysis of the L segment nucleotide sequences revealed 0.0–3.1% divergence among the Trans-Kama strains within each group (Figure 6, Appendix A). Even higher divergence, ranging from 1.2 to 9.1%, was found between groups. The nucleotide sequence comparison of the Trans-Kama strains demonstrated 2.5–9.5% of nucleotide sequence differences from Pre-Kama strains, as well as from Samara_49/CG/2005 and Puu/Kazan, which belong to RUS genetic lineage. In contrast, the Trans-Kama sequences displayed higher differences (14.7–16.7%) when compared with Bashkiria strains, which also belong to the RUS lineage. Likewise, Trans-Kama strains had 19.7–26.7% nucleotide sequence difference from FIN, CE, and N-SCA genetic lineages of PUUV (Figure 3, Appendix A). Previously, Ali et al. observed 17.0–19.0% diversity between PUUV strains that belong to the different genetic lineage in Central Europe, which is slightly lower than what was found in the current study [21]. A similar nucleotide sequence difference (from 17.6 to 24.7%) was found between Pre-Kama strains and strains from other PUUV lineages [19].

Similar to that observed in the N protein and glycoprotein precursor, analysis of the RdRp aa sequences of strains within the groups showed 0.0–0.5% differences. When these aa sequences were compared between the groups, the divergence ranged from zero to 2.3% (Appendix A). The aa sequence differences between the Trans-Kama strains and Pre-Kama strains as well as other RUS lineage strains were 0.0–1.8% and 0.5–2.3%, respectively, while corresponding differences were more noticeable (1.4–2.8%) when compared with Bashkiria strains. Further, higher differences (5.6–11.5%) were found when compared with the PUUV RdRp aa sequences of the different genetic lineages (Appendix A). We identified eight aa substitutions specific to Trans-Kama strains in individual groups: N346S and V466I were specific for group 1; K383R for groups 2a and 2b; while E347D, T419I, A436T, V460I, and I464V were found in groups 10, 4, 9, 3, and 2b, respectively (Figure 4). Interestingly, these specific mutations in Trans-Kama strains were not detected in any previously identified PUUV strain from the Pre-Kama area of the RT or in other RUS lineage strains.

### 3.3. Phylogenetic Analysis

The phylogenetic trees inferred from the PUUV S segment complete CDS sequences (1302 bp) and the partial M (1014 bp, nt 1499-2512) and L (665 bp, nt 958-1622) segments demonstrated similar topologies (Figure 7, Figure 8 and Figure 9, respectively). Trans-Kama PUUV strains formed two subclades, A and B. Both corresponded to the location of the certain bank vole trapping sites in the Trans-Kama area: subclades A and B, representing East and West, respectively. All three trees of subclade A contain strains from sites S1–S14 and are grouped into two branches A1 and A2. Subclade B had strains from sites S15–S21 and also formed two subclades (B1 and B2) on the phylogenetic trees. Interestingly, the position of strain MG_1992 (from site S10; denoted by a red circle on Figure 7, Figure 8 and Figure 9) on the S, M, and L phylogenetic trees differs. On the S segment tree, the strain MG_1992 formed its own branch clustering with the sequences from the sites S8, S9, and S11–S14 in subclade A1 (Figure 7). However, on the M segment tree, this strain is clustered outside of clades A and B, clustering with strains from Mamadysh and Sotyi of the Pre-Kama area recently classified as group C [19]. On the L segment tree strain MG_1992 formed an independent branch, which belongs to neither clade A nor clade B (Figure 9).

Similarly, there is different positioning of the strain MG_1555 on the S, M, and L phylogenetic trees (green circle on Figure 7, Figure 8 and Figure 9). On the trees for S and M segments, this strain is placed in the same subclade with the sequences from the sites S5 and S6 labeled N-K. On the contrary, on the L segment tree, the MG_1555 strain from site S6 is not grouped with strains from sites S5 and S6 (Figure 9). These inconsistencies in positioning of these strains suggest that this PUUV genome may be chimeric, consisting of the genome parts of different origins.

## 4. Discussion

In our previous study, we showed a significant diversity of the PUUV genomes in the Pre-Kama area of the RT [19]. It was demonstrated that the genetic distance between these orthohantavirus genome variants did not always correlate with the geographic distance between their location, with some PUUV sequences being more closely related to sequences from distant regions. We suggested that the observed distribution of PUUV genome variants is the result of a multidirectional migration of the bank voles, which came mainly from the south of RT. In this study, we investigated the distribution of PUUV genome variants in the Trans-Kama area located on the left bank of the Kama River and limited on the west by the Volga River (Figure 1).

In the current investigation, we have found that all the PUUV strains sequenced are grouped into the RUS clade on the phylogenetic trees based on the S segment CDS, and partial M and L segment sequences. In this clade, all investigated nucleotide sequences formed two subclades, A and B, corresponding to two areas of the Trans-Kama region: subclade A contains strains from the east, while strains in subclade B are distributed in the western areas of Trans-Kama (Figure 7, Figure 8 and Figure 9).

Interestingly, the composition and branch topologies within subclade B are almost identical on all three phylogenetic trees (S, M, and L segments). On all trees, nucleotide sequences of the MG_1041 strain from Teteevo found in the geographically distant region in the Pre-Kama area clustered with the Trans-Kama strains in subclade B2. Thus, one can suggest that the PUUV strains were brought to the Teteevo area as a result of the bank voles’ movement from the West Trans-Kama. It is believed that rivers act as an obstacle preventing rodent migration into the new areas of habitat. Our data suggest that obstacles, including rivers, even as wide as the Kama River, could be overcome under favorable circumstances. One of the possible favorable factors may be anthropogenic, as demonstrated by the bank vole invasion of Ireland in the 1920’s [22]. Similar anthropogenic factors that could contribute to the migration of the bank vole in the Teteevo area could be (i) cargo transportation along the Volga and Kama rivers for more than 1000 years; (ii) the presence of a bridge over the Kama River 20–25 km south of Teteevo until the mid-20th century, before the construction of the Kuibyshev dam and the formation of the Kuibyshev reservoir.

On all three phylogenetic trees, subclade A contained two subclades (A1 and A2). Subclade A1 includes strains circulating in the Zay River valley and in the adjacent territory (groups 4, 5, 6, 7). Subclade A2 strains come from the left bank of Kama River, upstream of the Kama River and Zay River confluence (groups 1, 2, 3). The tree topologies for S and M segments are almost identical (Figure 7 and Figure 8). The genetic distance between subclades A1 and A2 strains are in the range of 3.9–5.1% and 3.6–5.1%, respectively, for the S and M segment sequences, while within each subclade, these values do not exceed 3.1% (Appendix A). On the contrary, the tree topology for the L segment differs considerably from that of the S and M segments in the position of the branches comprising strains from groups 3 (subclade N-K) and 4 (subclade Z) (Figure 9).

The main difference in the phylogenetic tree topology was resultant of the MG_1992 strain (site S10, group 8), which was collected in the Kutemeli village. The S segment nucleotide sequence of this strain differs by 3.2% from the MG_1998 and MG_2001 strains from the closest trapping site (site S11, group 5) (Appendix A). The difference ranged from 3.1% to 3.7% when compared with strains from groups 4, 6, and 7. On the phylogenetic tree, the position of the strain MG_1992 branch indicates a close relationship with the strains of subclade A1 (Figure 7). Further, the M segment of MG_1992 strain has a relatively low nucleotide sequence divergence (Appendix A) compared with the strains from Mamadysh and Sotyi in the Pre-Kama area of the RT [19]. On the M segment tree, MG_1992 strain was grouped with the strains from the Pre-Kama area included in group C and distant from MG_1998 and MG_2001 branch (Figure 8). In contrast, strain MG_1992 formed a separate subclade, which was close to subclade A on the L segment phylogenetic tree (Figure 9), and its nucleotide sequence differed from the strains from the eastern part of the Trans-Kama area by 4.2–6.1% (Appendix A). These inconsistencies in positioning of strain MG_1992 sequences on phylogenetic trees suggest that this virus genome may contain parts of different ancestral origins. These could be a result of either whole-genome segment reassortment or RNA recombination. On the other hand, the S and L segments may have originated from more closely related genome variants than the M segment. This finding is consistent with observations made in Finland, which strongly suggested that some PUUV strains obtained from the bank vole are the result of segment reassortment between different genetic variants of the FIN PUUV lineage [23,24].

Certain genomic features of the MG_1992 strain appear to be associated with its geographical location and migration of bank voles, which could have been introduced into the Kutemeli area from different directions. It appears that the colonization of the Volga region by bank voles during the postglacial period occurred upstream of the Volga River and its tributaries (such as Kama River) from the southern refugium near the Sea of Azov [25,26]. Kutemeli village is located on a tributary of the Menzelya River (in turn, a tributary of the Kama River) near the watershed, which separates the basin of this river from the Zay River basin. It appears that bank voles were introduced into the Trans-Kama area by migrating upstream from the river valleys. In this scenario, we hypothesize that strains in groups 1, 2, and 3 (subclade A2) were introduced from the Samara region to the Nizhnekamsk and Naberezhnye Chelny areas through bank voles’ migration along the left banks of the Volga and Kama rivers (Figure 10). An indirect proof in favor of this hypothesis can be the degree of the PUUV strains’ divergence in the Trans-Kama area when compared with the Samara_49/CG/2005 strain. In most cases, along the bank of Kama River, these values rise with increasing distance from the Samara region. At the same time, we have found that the divergences increase when moving inland away from the Volga and Kama Rivers (Figure 10, Appendix A).

PUUV strains in groups 4, 5, 6, and 7 (subclade A1) could have been introduced into these areas through the north–south route of bank voles’ migration along the Zay River valley. In contrast, PUUV strains in the Kutemeli village could have spread as a result of northeast-to-southwest bank vole migration upstream of the Menzelya River basin. This hypothesis could be supported by the presence of specific mutations in the PUUV CDS S segment along the possible migration route of bank voles. These mutations include the R242 aa residue, which is encoded by AGG codon in all the subclade B strains from the West part of Trans-Kama area (except for MG_1419 strain); subclade A2 strains from the East; and also Teteevo and Samara strains (Figure 4). In the strains from subclade A1 from the West of Trans-Kama area, distributed in the Zay River valley, the R242 residue is encoded by the AGA codon. It is possible that the G>A mutation occurred in the ancestor PUUV strain during the bank voles’ movement in the Zay River valley from the bank of Kama River.

In the MG_1419 and MG_1992 strains, the K aa residue encoded by the AAA codon is located at position 242. It appears that mutation in these strains occurred independently. This codon in the ancestor of the MG_1419 strain could appear during the spread of bank voles upstream in the valley of Sheshma River. In contrast, in the ancestor of the strain MG_1992, the second G > A mutation occurred during the spread of bank voles upstream of the Menzelya River valley. Additional evidence, including corresponding sequences of additional strains circulating in Sheshma River and Menzelya River valleys with similar changes in the genome, is required to confirm our assumption.

It seems likely that the appearance of PUUV genome variants with unusual pathogenic potential in the RT could depend on the location of bank vole populations and geographical factors limiting contacts. In populations that have been constantly formed along bank voles’ migration routes—for example, in river valleys—the PUUV strains had limited genetic diversity. There, new genome variants could arise mainly as a result of nucleotide substitutions, genetic recombinations, and reassortment events, leading to limited deviation from the ancestral forms. Thus, the appearance of PUUV strains with new properties could only be possible through a relatively long evolution. Contrarily, in bank vole populations formed on different migration routes, the independent evolution of PUUV led to an increase in the genetic distance between strains. The particular geographical features of the habitat areas of individual populations, such as watersheds, could significantly reduce contact between neighboring populations and exchange of genetic materials between PUUV strains. However, in the periods of high bank vole population growth occurring every 3–4 years [27,28], contacts between individual rodents from adjacent populations become more frequent in some regions of watershed leading to an exchange of genetic material. In this case, different bank vole populations could form contact zones facilitating the exchange of genetic materials between diverged PUUV strains. This exchange may result in the emergence of recombinant and reassortant variants of PUUV genomic segments, with possible changes in infectivity to humans.

## Figures and Tables

**Figure 1 pathogens-10-01169-f001:**
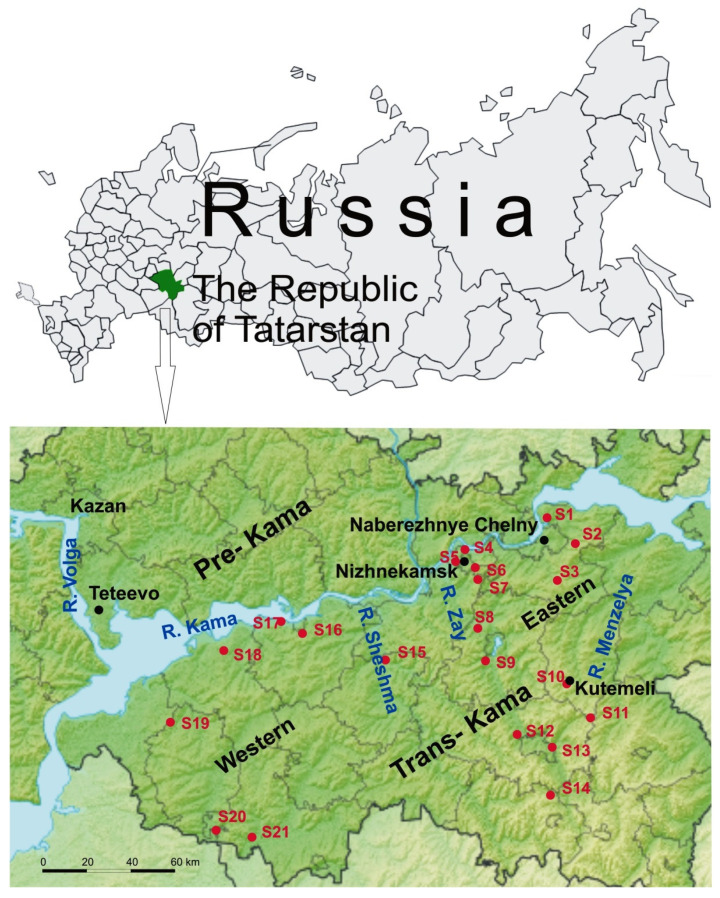
Geographic locations of the trapping sites (S1–S21) in the Trans-Kama area of the RT. The light blue areas show rivers (Volga and Kama) and red dots with a number representing the trapping sites. The map was modified from “Victor V”—Outline Map of Tatarstan.svgSRTM3, Public domain, from https://commons.wikimedia.org/w/index.php?curid=10983600 (accessed on 15 July 2021).

**Figure 2 pathogens-10-01169-f002:**
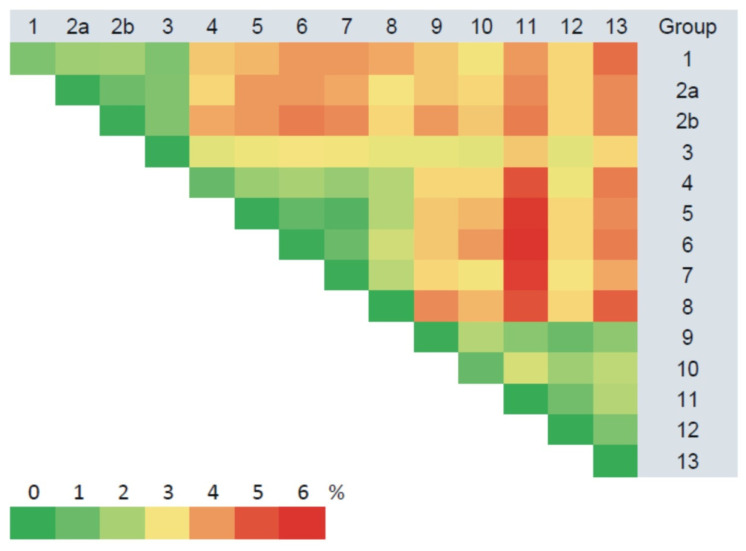
Divergence of the PUUV S segment nucleotide sequences among the PUUV strain genetic groups in the Trans-Kama area (%).

**Figure 3 pathogens-10-01169-f003:**
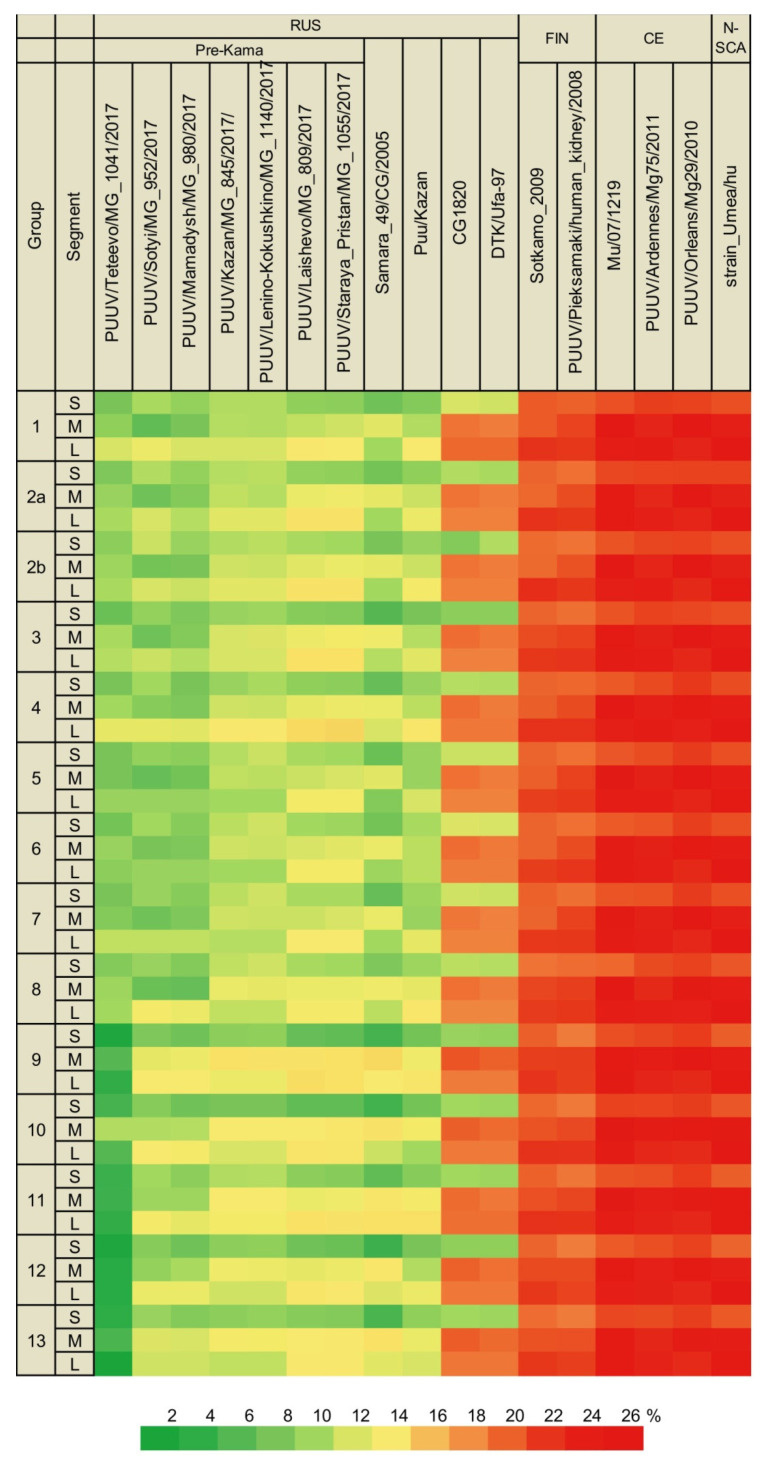
Nucleotide sequence divergence of the Trans-Kama PUUV strains compared with the Pre-Kama strains and strains of the RUS, FIN, CE, and N-SCA genetic lineages.

**Figure 4 pathogens-10-01169-f004:**
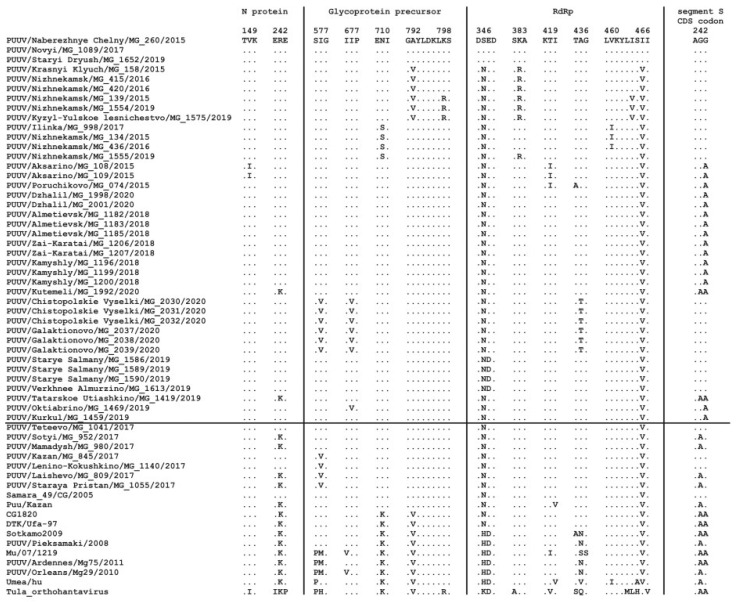
The identified aa substitutions in the CDS N protein, glycoprotein precursor and RdRp, and the position of codon 242 in the CDS sequence of the S segment.

**Figure 5 pathogens-10-01169-f005:**
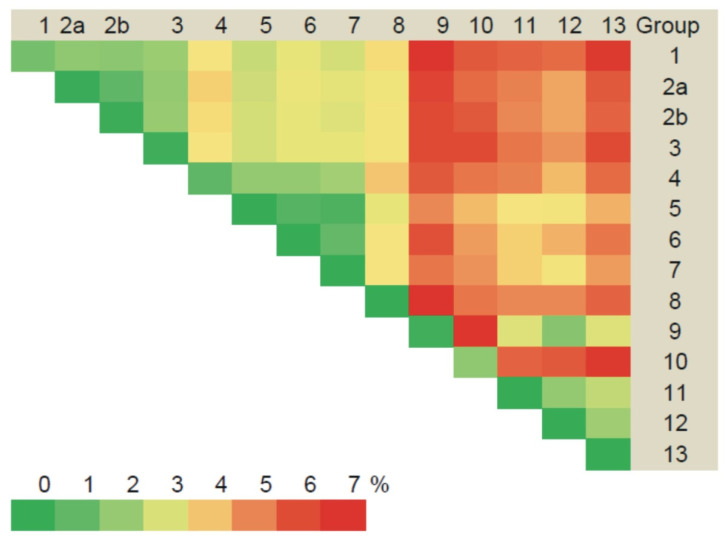
Divergence of the PUUV M segment nucleotide sequences among the PUUV strain genetic groups in the Trans-Kama area (%).

**Figure 6 pathogens-10-01169-f006:**
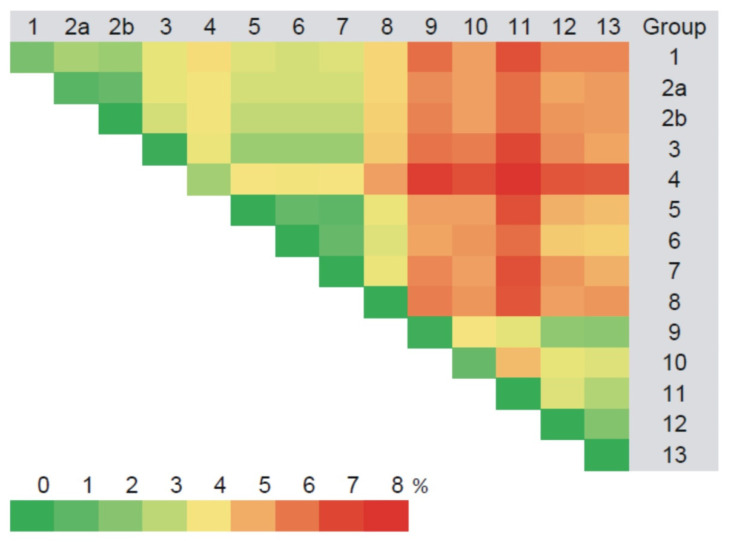
Divergence of the PUUV L segment nucleotide sequences among the PUUV strain genetic groups in the Trans-Kama area (%).

**Figure 7 pathogens-10-01169-f007:**
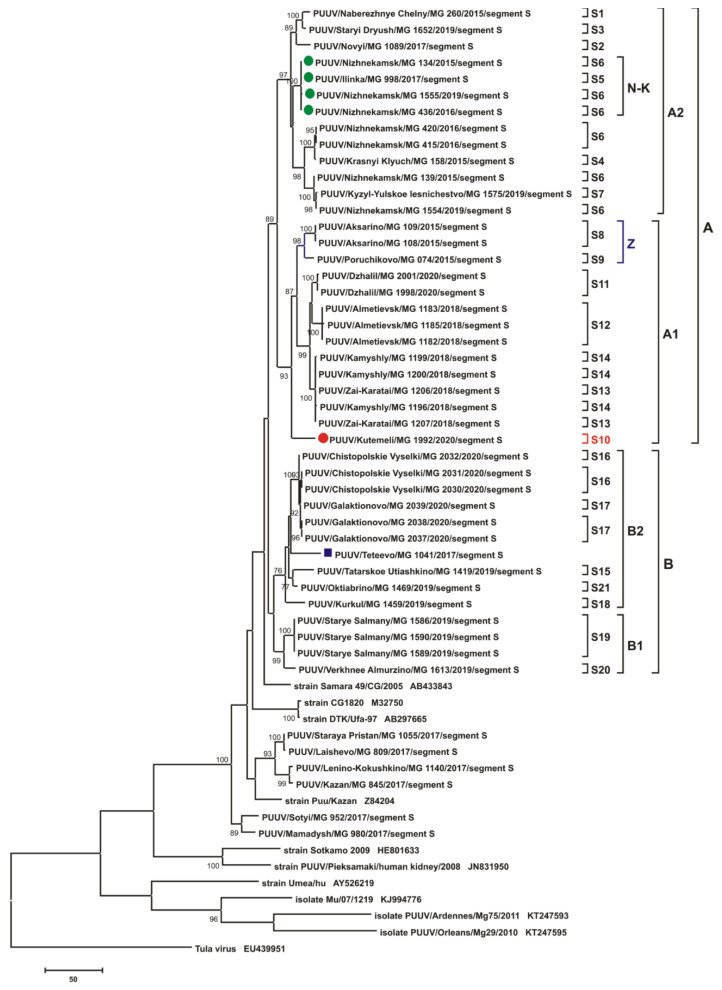
The phylogenetic tree based on the S segment complete CDS nucleotide sequences of PUUV Trans-Kama and reference strains. Phylogenetic tree was generated using complete CDS of S segment nucleotide sequences (1302 bp-long, nt 43-1344). S segment nucleotide sequence positions were numbered based on GenBank sequence Kazan, accession number Z84204. Maximum Parsimony method was used for the analysis. The percentages of replicate trees in which the certain taxa clustered together in the bootstrap test (1000 replicates) are shown next to the corresponding branch nodes, and only values >70% are shown. The trapping sites where the corresponding strains were identified are indicated from S1 to S21, whilst the A, A1, A2, B, B1, B2, Z, and N-K indicate the subgroups formed. Dark blue square indicates strain Teteevo/MG_1041 from Pre-Kama area. The red circle indicates strain MG_1992, which changes its position on the different trees. The green circle indicates strains from subclade N-K, which change their positions on the different trees.

**Figure 8 pathogens-10-01169-f008:**
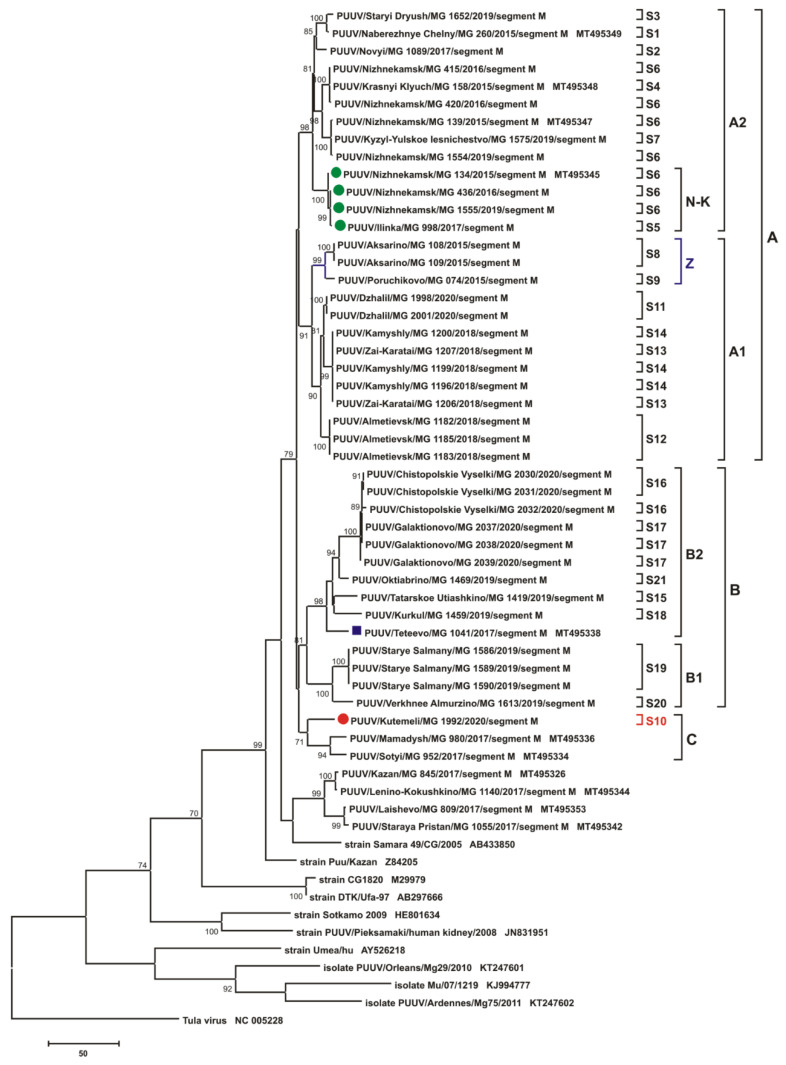
The phylogenetic tree based on the partial M segment nucleotide sequences of PUUV Trans-Kama and reference strains. Phylogenetic tree was generated using complete CDS of S segment nucleotide sequences (1014 bp-long, nt 1499-2512). M segment nucleotide sequence positions were numbered based on GenBank sequence Kazan, accession number Z84205. Maximum Parsimony method was used for the analysis. The percentages of replicate trees in which the certain taxa clustered together in the bootstrap test (1000 replicates) are shown next to the corresponding branch nodes and only values >70% are shown. The trapping sites where the corresponding strains were identified are indicated from S1 to S21, whilst the A, A1, A2, B, B1, B2, Z, and N-K indicate the subgroups formed. The dark blue square indicates strain Teteevo/MG_1041 from Pre-Kama area. The red circle indicates strain MG_1992, which changes its position on the different trees. The green circle indicates strains from subclade N-K, which change their positions on the different trees.

**Figure 9 pathogens-10-01169-f009:**
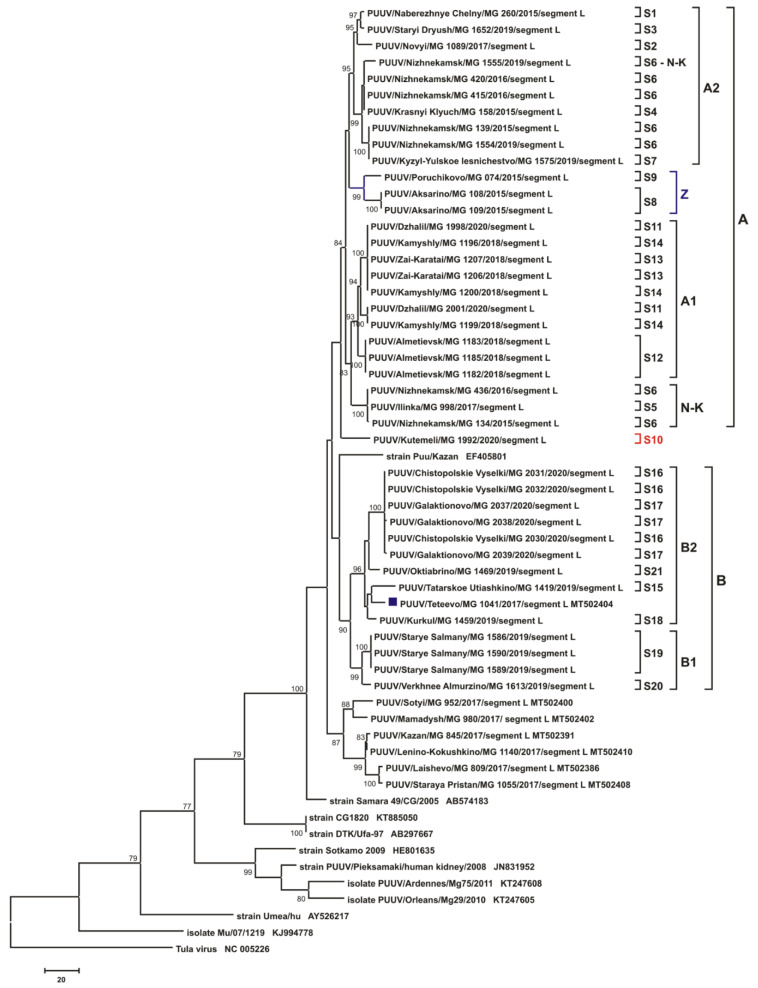
The phylogenetic tree based on the partial L segment nucleotide sequences of PUUV Trans-Kama and reference strains. Phylogenetic tree was generated using complete CDS of S segment nucleotide sequences (665 bp-long, nt 958-1622). L segment nucleotide sequence positions were numbered based on GenBank sequence Kazan, accession number EF405801. Maximum Parsimony method was used for the analysis. The percentages of replicate trees in which the certain taxa clustered together in the bootstrap test (1000 replicates) are shown next to the corresponding branch nodes, and only values >70% are shown. The trapping sites where the corresponding strains were identified are indicated from S1 to S21, whilst the A, A1, A2, B, B1, B2, Z, and N-K indicate the subgroups formed. The dark blue square indicates strain Teteevo/MG_1041 from Pre-Kama area. The red circle indicates strain MG_1992, which changes its position on the different trees. The green circle indicates strains from subclade N-K, which change their positions on the different trees.

**Figure 10 pathogens-10-01169-f010:**
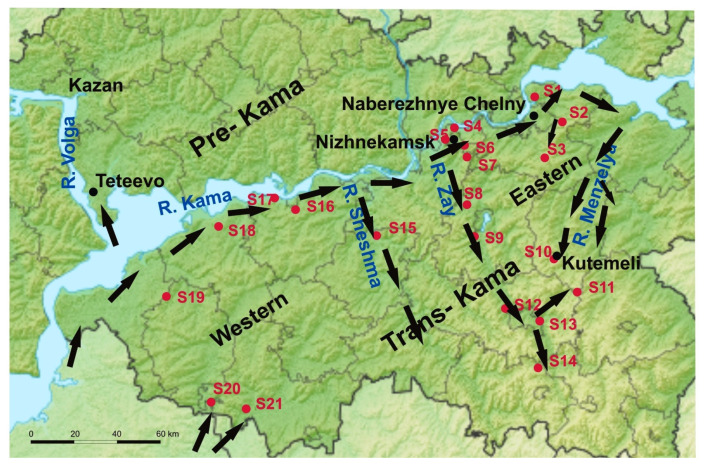
Possible migratory routes of bank voles in Trans-Kama area of the RT. The black arrows indicate the hypothetical movements of bank voles from the southern refugium. The map was modified from “Victor V”—Outline Map of Tatarstan.svgSRTM3, Public domain, from https://commons.wikimedia.org/w/index.php?curid=10983600 (accessed on 15 July 2021).

**Table 1 pathogens-10-01169-t001:** Bank vole trapping site locations, number of trapped individuals, and nucleotide sequences obtained from each location.

Trapping Site	Location	No. of Trapped Bank Voles	No. of RT-PCR Positive Voles	No. Sequences Used for Analysis (Segment)		
				S (1302) ^a^	M (1014) ^a^	L (665) ^a^
S1	Naberezhnye Chelny	10	3	1	1	1
S2	Novyi	17	5	1	1	1
S3	Staryi Dryush	7	2	1	1	1
S4	Krasnyi Klyuch	18	5	3	3	3
S5	Ilinka	15	2	1	1	1
S6	Nizhnekamsk	21	8	5	5	5
S7	Kyzyl-Yulskoe lesnichestvo	10	1	1	1	1
S8	Aksarino	3	3	2	2	2
S9	Poruchikovo	5	5	1	1	1
S10	Kutemeli	5	1	1	1	1
S11	Dzhalil	5	2	2	2	2
S12	Almetievsk	4	3	3	3	3
S13	Zai-Karatai	4	3	2	2	2
S14	Kamyshly	6	3	3	3	3
S15	Tatarskoe Utiashkino	1	1	1	1	1
S16	Chistopolskie Vyselki	5	3	3	3	3
S17	Galaktionovo	5	3	3	3	3
S18	Kurkul	3	1	1	1	1
S19	Starye Salmany	8	3	3	3	3
S20	Verkhnee Almurzino	8	1	1	1	1
S21	Oktiabrino	2	1	1	1	1
	Total	162	59	40	40	40

^a^ Numbering corresponds to the position of RT nucleotide sequences on the PUUV strain Puu/Kazan used for comparison (Genbank Accession No Z84204, Z84205, and EF405801 for S, M, and L segment, respectively).

## Data Availability

Not applicable.

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
