# Peer review of "The Distribution of Puumala orthohantavirus Genome Variants Correlates with the Regional Landscapes in the Trans-Kama Area of the Republic of Tatarstan"

_pathogens, 2021, doi:10.3390/pathogens10091169_

Round 1

Reviewer 1 Report

Summary:

Davidyuk et al. report genomic comparisons of PUUV from RNA isolated from bank voles in the Trans-Kama region of the Republic of Tatarstan. From the generated data, the authors assessed the intra- and inter-group variability of viral genomes isolated from 21 different sites, as well as compared to previously published data. Based on the phylogenetic patterns of the S, M, and L segments that were sequenced, the authors inferred potential migration patterns and virus evolution.

Broad Comments:

  • The analyses are generally straightforward. However, the material is dense at times which can make it challenging to follow. In particular, the use of Tables as the primary way to report the data can make interpretation tedious for the first-time reader. Graphical/Figure representations would be beneficial to highlight the qualitative relationships (intra- vs inter-group variability, etc), with the Tables supplying detailed information.

Specific Comments:

  • Page 3, Figure 1: What was the source of the annotated maps? Was it created solely by the authors, or were they modified from another source?
  • Page 4, Lines 139-140: The wording is a little confusing—is it being stated that the CDS could be obtained for 40 out of 59 RT-PCR positive samples?
  • Page 5, Lines 155-156: Does the 6% value refer to divergence between different groups? If so, the best word would be “inter-group”
  • Page 5, Lines 156-159: Is this sentence discussing data in Table 3? If so, the in-text citation should be added.
  • Page 5, Line 64: Typo—“indentified” should be “identified”
  • Page 10, Lines 1-7; Page 12, Lines 1-13; Page 14, Lines 1-14: It would be helpful to include sequence alignments with residue numbering to highlight the amino acids discussed in these sections. While it need not be exhaustive, it would provide helpful context to grasp which part of the N, GPC, and L proteins contain the mutations.
  • Page 20, Lines 63-69: Could the variable positioning of these strains in the phylogenetic trees be an artifact of comparing partial, rather than complete, CDS of M and L?
  • Page 20, Lines 93-94: Are there any particular circumstances that would be considered “favorable”?

Author Response

The authors would like to express gratitude to the reviewers for their very constructive comments and detailed suggestions for the manuscript. The authors believe that the comments and recommendations offered by the reviewers have identified important areas which needed improvement. In this regard, the authors have revised the text, incorporated all the suggestions according to the two reviewers.

Summary:

Davidyuk et al. report genomic comparisons of PUUV from RNA isolated from bank voles in the Trans-Kama region of the Republic of Tatarstan. From the generated data, the authors assessed the intra- and inter-group variability of viral genomes isolated from 21 different sites, as well as compared to previously published data. Based on the phylogenetic patterns of the S, M, and L segments that were sequenced, the authors inferred potential migration patterns and virus evolution

The authors thank the reviewer for their kind words.

Broad Comments:

The analyses are generally straightforward. However, the material is dense at times which can make it challenging to follow. In particular, the use of Tables as the primary way to report the data can make interpretation tedious for the first-time reader. Graphical/Figure representations would be beneficial to highlight the qualitative relationships (intra- vs inter-group variability, etc), with the Tables supplying detailed information.

The authors thank the reviewer for his broad comments. We agree with reviewer’s proposition about refining the Tables. For better visualization, we have replaced the digital representation of information in the tables with colored figures and the original Tables 2, 3, 4, 5 are placed into the Supplementary data.

Specific Comments:

Page 3, Figure 1: What was the source of the annotated maps? Was it created solely by the authors, or were they modified from another source?

The authors apologize for not acknowledging the author of the map though it is in public domain for free use. However, the authors have modified the captions to Figures 1 and 10 to include the source of the original map. For example, the caption to Figure 1 now reads “Geographic locations of the trapping sites (S1–S21) in the Trans-Kama area of the RT. The light blue areas show rivers (Volga and Kama) and red dots with a number representing the trapping sites. The map was modified from “Victor V” – Outline Map of Tatarstan.svgSRTM3, Public domain, from https://commons.wikimedia.org/w/index.php?curid=10983600”

Page 4, Lines 139-140: The wording is a little confusing—is it being stated that the CDS could be obtained for 40 out of 59 RT-PCR positive samples?

The authors agree and the sentence was reworded.

Page 5, Lines 155-156: Does the 6% value refer to divergence between different groups? If so, the best word would be “inter-group”.

The authors have made the typographical change to “inter-group”.

Page 5, Lines 156-159: Is this sentence discussing data in Table 3? If so, the in-text citation should be added.

The authors agree that they missed to add the citation of table 3 in the text. Table 3 was added in the text.

Page 5, Line 64: Typo—“indentified” should be “identified”

The authors have made the typographical change

Page 10, Lines 1-7; Page 12, Lines 1-13; Page 14, Lines 1-14: It would be helpful to include sequence alignments with residue numbering to highlight the amino acids discussed in these sections. While it need not be exhaustive, it would provide helpful context to grasp which part of the N, GPC, and L proteins contain the mutations.

We agree with reviewer’s proposition and have added a figure which shows the identified aa substitutions and codon 242 in CDS sequence of the S segment. This is now labelled as Figure 4.

Page 20, Lines 63-69: Could the variable positioning of these strains in the phylogenetic trees be an artifact of comparing partial, rather than complete, CDS of M and L?

The authors agree that it is impossible to base the interpretation of the data on one fragment of the genome segment. However, the variability of the CDS of the M and L segments do not significantly differ [Ali et al, 2015]. Therefore, we assume that the phylogenetic trees constructed using longer genome fragment will have a similar topology to the one constructed using short fragment. At the same time, the authors agree that the presence of recombinant sites in the genome segments can change the topology of the trees.

Page 20, Lines 93-94: Are there any particular circumstances that would be considered “favorable”?

The authors thank the reviewer for the constructive question. The explanation was added to the appropriate paragraph in the manuscript:

One of the possible favorable factors may be anthropogenic, as it was demonstrated by bank vole invasion of the island of Ireland in the 1920’s (White et al, 2012). Similar anthropogenic factors that could contribute to the migration of the bank vole in the Teteevo area could be: (i) cargo transportation along the Volga and Kama rivers for more than 1000 years; (ii) the presence of a bridge over the Kama River 20-25 km south of Teteevo until the mid-20th century, before the construction of the Kuibyshev dam and the formation of the Kuibyshev reservoir.”

Reviewer 2 Report

Line 100. How were these bank voles captured? How were tissues taken and stored until processing?

Figures 1 and 5. It would be helpful to include a distance (km) ruler line at the bottom of the maps to know the distances between the trapping sites.

Discussion line 93. What types of circumstances? Suggest some. 

Author Response

The authors would like to express gratitude to the reviewers for their very constructive comments and detailed suggestions for the manuscript. The authors believe that the comments and recommendations offered by the reviewers have identified important areas which needed improvement. In this regard, the authors have revised the text, incorporated all the suggestions according to the two reviewers.

Comments and Suggestions for Authors

Line 100. How were these bank voles captured? How were tissues taken and stored until processing?

The authors thank the reviewer for the comment. The authors obtained frozen rodent lung tissue samples from Federal Healthcare Institute "Centre for Hygiene and Epidemiology in the Republic of Tatarstan (Tatarstan)", which is under Federal Service for Surveillance on Consumer Rights Protection and Human Wellbeing. This institute is responsible for epidemiological monitoring of HFRS in the Republic of Tatarstan and they trap and store biomaterials according to the international regulations.

Figures 1 and 5. It would be helpful to include a distance (km) ruler line at the bottom of the maps to know the distances between the trapping sites.

The authors agree and the scale was added to Fig. 1 and Fig. 10.

Discussion line 93. What types of circumstances? Suggest some.

The authors thank the reviewer for the constructive suggestion. The explanation was added to the appropriate paragraph in the manuscript:

"One of the possible favorable factors may be anthropogenic, as it was demonstrated by bank vole invasion of the island of Ireland in the 1920’s (White et al, 2012). Similar anthropogenic factors that could contribute to the migration of the bank vole in the Teteevo area could be: (i) cargo transportation along the Volga and Kama rivers for more than 1000 years; (ii) the presence of a bridge over the Kama River 20-25 km south of Teteevo until the mid-20th century, before the construction of the Kuibyshev dam and the formation of the Kuibyshev reservoir."